https://doi.org/10.1038/s41467-021-21485-w | **OPEN**

# Data linkage to evaluate the long-term risk of HIV infection in individuals seeking post-exposure prophylaxis

Frédérique Hovaguimian [1,2,3 ✉], Huldrych F. Günthard [1], Christoph Hauser[4], Anna Conen [5], Enos Bernasconi [6], Alexandra Calmy[7], Matthias Cavassini[8], Marco Seneghini[9], Alex Marzel[10], Henriette Heinrich[11,12], Alexandra Scherrer[1], Julien Riou[13], Adrian Spoerri[13], Kurt Schmidlin[13], Suraj Balakrishna[1,3], Dominique L. Braun[1], Silvana K. Rampini [14,31], Jan S. Fehr[1,2,31], Roger D. Kouyos [1,3,31], the Swiss HIV Cohort Study*

Evidence on the long-term risk of HIV infection in individuals taking HIV post-exposure prophylaxis remains limited. In this retrospective data linkage study, we evaluate the occurrence of HIV infection in 975 individuals who sought post-exposure prophylaxis in a tertiary hospital between 2007 and 2013. Using privacy preserving probabilistic linkage, we link these 975 records with two observational databases providing data on HIV events (Zurich Primary HIV Infection study and the Swiss HIV Cohort Study). This enables us to identify 22 HIV infections and to obtain long-term follow-up data, which reveal a median of 4.1 years between consultation for post-exposure prophylaxis and HIV diagnosis. Even though men who have sex with men constitute only 35.8% of those seeking post-exposure prophylaxis, all 22 events occur in this subgroup. These findings should strongly encourage early consideration of pre-exposure prophylaxis in men who have sex with men after a first episode of post-exposure prophylaxis.

[1] Division of Infectious Diseases and Hospital Epidemiology, University Hospital of Zurich, Zurich, Switzerland. [2] Department of Public and Global Health, Epidemiology, Biostatistics and Prevention Institute, University of Zurich, Zurich, Switzerland. [3] Institute of Medical Virology, University of Zurich, Zurich, Switzerland. [4] Department of Infectious Diseases, Bern University Hospital, University of Bern, Bern, Switzerland. [5] Department of Infectious Diseases and Hospital Hygiene, Kantonsspital Aarau, Aarau, Switzerland. [6] Division of Infectious Diseases, Regional Hospital Lugano, Lugano, Switzerland. [7] Laboratory of Virology and Division of Infectious Diseases, Geneva University Hospital, Geneva, Switzerland. [8] Division of Infectious Diseases, Lausanne University Hospital, Lausanne, Switzerland. [9] Division of Infectious Diseases and Hospital Epidemiology, Kantonsspital St. Gallen, St. Gallen, Switzerland. [10] Research, Teaching and Development, Schulthess Clinic, Zurich, Switzerland. [11] Department of Gastroenterology and Hepatology, University Hospital of Zurich, Zurich, Switzerland. [12] Department of Gastroenterology, Stadtspital Triemli, Zurich, Switzerland. [13] Institute of Social and Preventive Medicine, University of Bern, Bern, Switzerland. [14] Division of Internal Medicine, University Hospital of Zurich, University of Zurich, Zurich, Switzerland. [31]These authors contributed equally: Silvana K. Rampini, Jan S. Fehr, Roger D. Kouyos. *A list of authors and their affiliations appears at the end of the paper. ✉email: frederique.lachmann@usz.ch

Post-exposure prophylaxis (PEP) is a widely accepted measure to prevent HIV transmission[1], and has been an integral part of Switzerland's prevention program since 1997[2]. Although indication criteria for non-occupational PEP and drug regimens may slightly differ across healthcare institutions, PEP is commonly prescribed to individuals within 48–72 h after unprotected anal or vaginal sexual exposure, with a HIV-positive partner not on suppressive antiretroviral therapy, or with a partner of unknown HIV-status belonging to groups at considerable risk of HIV transmission (i.e., men who have sex with men (MSM), sex workers, people using intravenous drugs, and individuals from a region with high HIV prevalence (defined as >10%))[3]. In individuals receiving PEP, 3-compound antiretroviral therapy is administered for 4 weeks, and follow-up should include testing for HIV at 3 months to exclude late seroconversions[2,3].

Whilst PEP intake in itself generally results in favorable outcome, some subgroups seem to remain at high risk for subsequent HIV infection[4,5]. To date, evidence on the long-term risk of HIV in PEP seekers remains limited and conflicting: HIV incidence after PEP intake was found to range from 0.78 to 7.6 per 100 person-years, but these findings were partly driven by small studies, conducted over short time periods (i.e., typically 6–12 months), and including exclusively MSM[4–12]. Although large cohort studies with prolonged follow-up are needed to better characterize populations with a higher long-term risk of HIV infection, loss to follow-up often undermines the feasibility of such studies.

One possible approach to mitigate loss to follow-up bias resides in the linkage of different health-related databases[13,14]. Linkage may be performed between different datasets using a common unique identifier (e.g., linkage of electronic records between two departments within a single institution), or using probabilistic record linkage whenever unique identifiers are not available (e.g., linkage of records from two independent cohort studies)[14]. Probabilistic linkage commonly uses variables considered as personal identifying information, such as names or date of birth. In recent years, linkage of personal identifying variables have become increasingly challenging due to privacy protection laws, thereby leading to the development of new methodological approaches[15]. Among others, the Privacy Preserving Probabilistic Record Linkage (P3RL) has been found to be a reliable method to preserve confidentiality: after one-way data encryption by the responsible datacenters, a third, independent party is involved to perform probabilistic data linkage[15].

The primary aim of this retrospective data linkage study was to estimate the long-term risk of HIV infection in PEP seekers. More specifically, we assessed the occurrence of HIV diagnoses until October 2019 in individuals who sought PEP at a tertiary referral hospital in Switzerland (University Hospital of Zurich—USZ) between 2007 and 2013. To achieve this, we used a privacy preserving data linkage method between 3 different databases. As previous evidence suggests that sexual risk taking occurs in phases lasting 12 to 24 months[16], we hypothesized that HIV infections would be clustered shortly after the time of PEP consultation. We were also interested in characterizing subgroups at higher risk of HIV infection, i.e., to assess their clinical presentation and to explore which factors were associated with long-term risk of HIV diagnosis. Additional analyses aimed at assessing whether the decision to prescribe PEP at the time of PEP consultation was appropriate.

## Results

**Data linkage.** Of 975 records included in the PEP-USZ database, 15 were identified as potential links in the USZ-specific part of the SHCS or in the ZPHI Study (Fig. 1). Of these, 3 were further excluded since these individuals were diagnosed with HIV at PEP consultation. Thus, internal linkage retrieved 12 records of PEP seekers diagnosed with HIV.

Probabilistic record linkage retrieved 34 potential pairs. Of these, we excluded 10 further records: 9 were potential links, which—after further assessment of patient chart with the responsible centers—did not constitute true matches, and in 1 individual, HIV was diagnosed at PEP consultation. Of the remaining 24 potential links, 14 were further discarded, since these records were already identified through internal linkage. Thus, 10

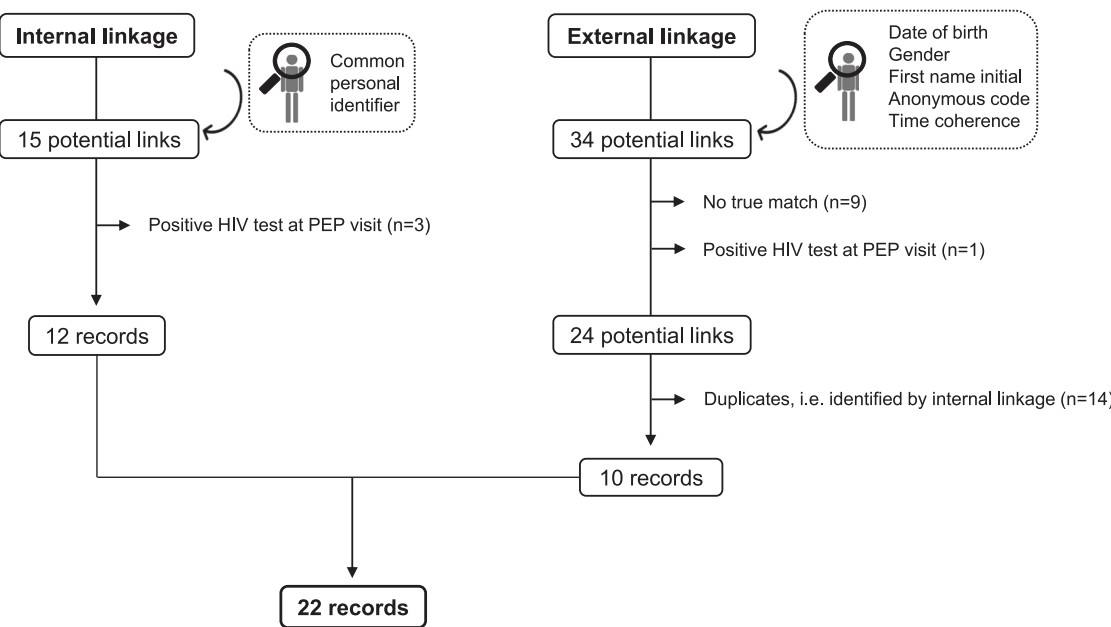

**Fig. 1 Flow diagram illustrating the data linkage process.** Of 975 records included in the PEP-USZ database, internal linkage yielded 12 records. Ten additional records were retrieved through external linkage. Thus, data linkage resulted in the identification of 22 HIV infections among individuals seeking PEP between 2007 and 2013. PEP: post-exposure prophylaxis; USZ: university hospital of Zurich.

additional verified records of PEP seekers diagnosed with HIV were retrieved using probabilistic linkage with the SHCS.

**Occurrence of HIV infections and characteristics of PEP seekers.** Combining records from internal and external linkages resulted in the identification of 22 HIV infections: all 22 links were participants enrolled in the SHCS (12 USZ-specific, 10 non-USZ), of which 7 were also included in the ZPHI. Evidence of HIV infection was thus found in 2.3% (22/971, 95%CI 1.48 to 3.43) of the initial PEP seekers cohort. Characteristics of individuals included in the cohort are presented in Table 1.

Compared to PEP seekers without evidence of HIV infection, those diagnosed with HIV were all MSM (100% versus 34.4%, p-value = <0.0001—Table 1). Although not reaching statistical significance, repeated PEP intake, either in the past or during study duration, differed between HIV outcome status. When only MSM were considered, the proportion of individuals with

evidence of HIV infection reached 6.3% (22/348, 95%CI 4.17 to 9.43).

**Clinical trajectories of PEP seekers diagnosed with HIV.** Figure 2 illustrates the clinical trajectories of PEP seekers diagnosed with HIV (n = 22). The median time between the last PEP consultation and HIV diagnosis was 4.1 years (IQR: 2.3–6.4). In most cases, the possible infection window was shorter than 2 years: the median time between the earliest possible infection date and HIV diagnosis was 6.4 months (IQR: 2.1–20.8) (Fig. 3—panel a). All 22 individuals were diagnosed in Zurich. Of these, 10 HIV diagnoses were made outside the USZ (private practice setting).

In-depth review of the SHCS and ZPHI data revealed that 9 individuals (40.9%) presented clinical signs compatible with primary HIV infection (i.e., acute or recent infection): 7 were enrolled in the ZPHI Study, and 2 presented a time difference between negative and positive HIV tests of less than 90 days

**Table 1 Characteristics of individuals seeking PEP at the University Hospital of Zurich between 2007 and 2013.**

|  | Overall n = 971[a] | Presumed HIV negative n = 949 | HIV positive n = 22 | p-value |
|---|---|---|---|---|
| Age[b], median [IQR] | 32 [26 – 38] | 32 [26 – 38] | 32 [26 – 40] | 0.745 |
| Sex, male, n (%) | 797 (82.1) | 775 (81.7) | 22 (100) | 0.021 |
| Swiss nationality, n (%) | 684 (70.4) | 670 (70.6) | 14 (63.6) | 0.483 |
| MSM, n (%) | 348 (35.8) | 326 (34.4) | 22 (100) | <0.0001 |
| Repeated PEP, n (%) | 96 (9.9) | 92 (9.7) | 4 (18.2) | 0.263 |
| PEP indicated, n (%) | 528 (54.4) | 510 (53.7) | 18 (81.8) | 0.009 |

[a]Records from individuals diagnosed with HIV at PEP visit (n = 4) were excluded from original database (n = 975), as described in Fig. 2.
[b]Age recorded at initial presentation (i.e., first PEP consultation)
PEP seekers without HIV diagnosis (n = 949) were compared to those with evidence of HIV infection (n = 22). The p values were obtained using the Man–Whitney U test for continuous variables and the Fisher exact test for binary variables (two-sided tests). No adjustment for multiple comparisons was made (main study aim: descriptive). MSM: men who have sex with men; PEP: post-exposure prophylaxis.

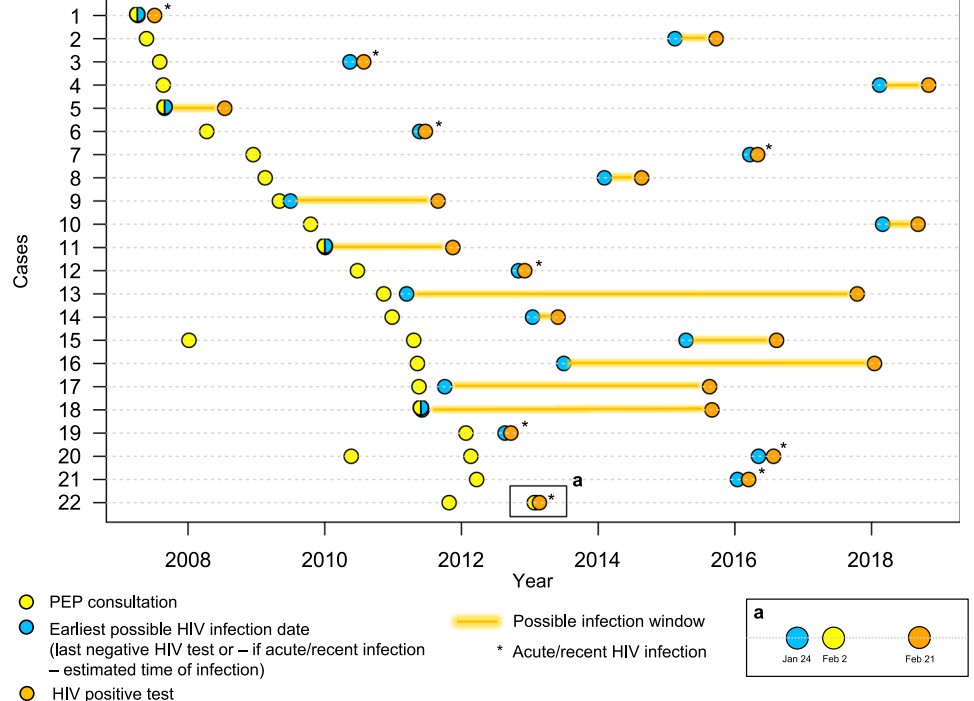

**Fig. 2 Clinical trajectories of PEP seekers diagnosed with HIV (n = 22).** Nine individuals presented clinical signs compatible with primary HIV infection (i.e., acute or recent infection). PEP failure was possible in 6 cases: the start of the infection window occurred within 3 months of PEP consultation (cases number 5, 9, 11, 18 and 22), or HIV was diagnosed at 3 months (case number 1). In case number 22, the estimated time of infection and medical history were compatible with transmission 9 days before PEP consultation (part a). PEP: post-exposure prophylaxis.

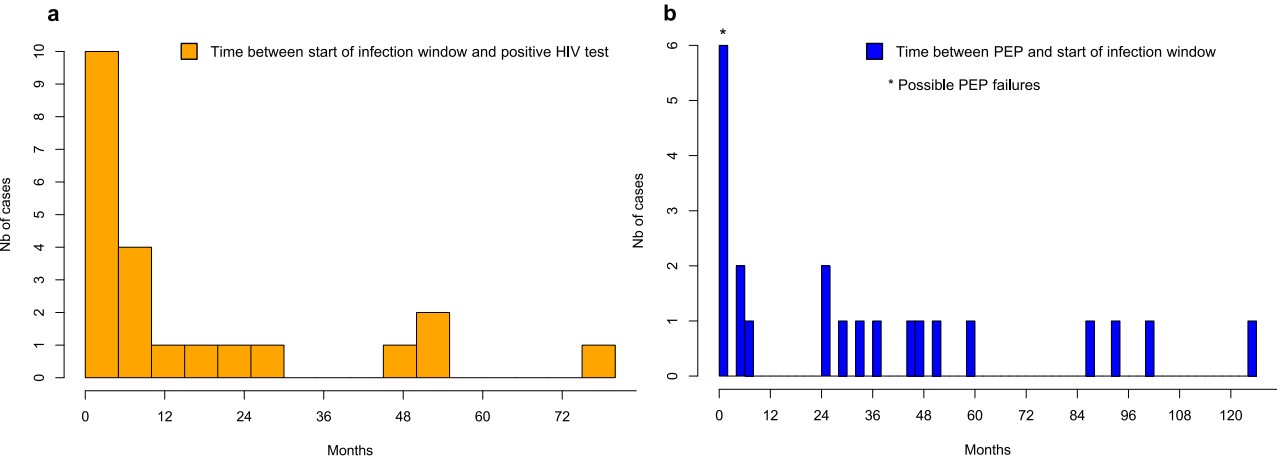

**Fig. 3 Infection window and possible PEP failures: distribution among PEP seekers diagnosed with HIV (*n* = 22). a** Histogram showing the distribution of time between the earliest possible infection date and HIV diagnosis (orange bars). In 17 cases, the possible infection window was shorter than 2 years. **b** Histogram showing the distribution of time between PEP consultation and the start of the infection window (dark blue bars). PEP failure was possible in 6 cases (illustrated with *): the start of the infection window occurred within 3 months of PEP consultation (*n* = 5), or HIV was diagnosed at 3 months (*n* = 1). PEP: post-exposure prophylaxis.

(Fig. 2). In 6 others (27.3%), the sequence of events did not allow to rule out PEP failure: in 5 individuals, the start of the infection window occurred within 3 months of PEP consultation (i.e., infection possibly related to the event that triggered PEP—Fig. 2: individuals number 5, 9, 11, 18 and 22, Fig. 3—panel b), and in individual number 1, the HIV test at 3 months was found positive. Detailed data review, however, revealed that individual number 18 did not receive PEP (reason: no indication criteria), and that in 2 individuals, there was evidence of repetitive sexual exposures, thereby making PEP failure unlikely (i.e., number 9 and 22). Interestingly, in individual number 22, the estimated time of infection and medical history were compatible with transmission 9 days before PEP consultation (Fig. 2, part a). In-depth review of the SHCS and ZPHI data did not identify PEP consultations other than those retrieved by our linkage.

**Factors associated with long-term risk of HIV diagnosis.** We aimed to explore whether specific factors were associated with a long-term risk of HIV: some risk factors, however, were so strong that they accounted for all outcome events (e.g., all HIV infections occurred in MSM), thereby impairing meaningful multivariable regression analysis. Other methodological approaches (i.e., firth logistic regression[17], which usually accounts for highly predictive risk factors) did not yield meaningful results, as some degree of data separation was still present.

We also assessed the validity of the PEP prescription algorithm: compared to PEP seekers without evidence of HIV infection, those diagnosed with HIV were more likely to have an indication for PEP (81.8% in those with and 53.7% in those without evidence of HIV infection, *p*-value = 0.009—Table 1). Univariable regression analysis confirmed this association (OR 3.87, 95%CI 1.43 to 13.49) and results remained consistent after adjustment for repetitive PEP seeking (OR 3.72, 95%CI 1.37 to 12.99).

**Expected false-negative proportion yielded by data linkage.** To estimate the expected false-negative proportion yielded by our method, we performed a simulation of our linkage using two pre-existing registries. In this simulation study, 5.4% of false-negative pairs were expected.

**Discussion**

In this retrospective data linkage study including a large and diverse PEP seeker population, we found that HIV infections occurred several years after PEP consultation, i.e., much later than originally assumed. Even though MSM constituted only 35.8% of PEP seekers, infections were found exclusively in this subgroup. Clinical indicators, such as repetitive PEP seeking or indication for PEP, were associated with subsequent HIV infection.

According to evidence on sexual behavior, sexual risk taking in HIV-negative MSM seems to follow distinct patterns over time: in a longitudinal analysis of data from the Multicenter AIDS cohort Study[16], individuals deemed at moderate risk of seroconversion presented phases of risky behavior that lasted on average 12 months, whilst those deemed at high risk exhibited phases of 24 months. Thus, considering PEP seeking as a marker of sexual risk taking, we assumed that outcome events would be clustered 12 to 24 months after PEP seeking. Surprisingly, however, most HIV events in our study occurred several years after PEP consultation. Our study suggests thus that phases of risk may persist for longer periods than originally assumed, or that some trajectories may include recurrent phases of risk. Large cohort studies enrolling HIV-negative individuals at considerable risk of seroconversion— such as the SwissPrEPared cohort study (NCT03893188)—will bring a better understanding of sexual behavior and risk trajectories over time.

Repetitive PEP seeking has been widely regarded as an indication for the prescription of pre-exposure prophylaxis (PrEP)[18,19]. This is consistent with our findings, where repeated PEP intake, either in the past or during study duration, differed between HIV outcome status. In our study, however, HIV events were found to occur exclusively in MSM, which raises the additional question whether MSM should be encouraged to transition to PrEP as early as after a first PEP episode. Early transition is also supported by the fact that, compared to HIV incidence rates in the overall MSM population of Zurich (average 2010–2013: 39 per 10,000)[20], rates retrieved from our study were higher in PEP-seeking MSM (70.5 per 10,000) and in MSM with repetitive PEP seeking (81.1 per 10,000; assuming that all HIV events were captured, and that all participants presumed HIV negative were still at risk at the time of analysis—Supplementary table 1). Although some institutions

may have already started considering first PEP-episodes in MSM as an indication for PrEP (as it is currently the case at the USZ), the exact timing of referral to PrEP programs is still lacking in most national and international recommendations[18,21,22]. Findings from our study may thus further contribute to closing the knowledge gap regarding the timing of PEP to PrEP transition in MSM[22]. More specifically, a first, successful PEP episode (i.e., completed without subsequent HIV infection) should trigger eligibility screening for PrEP.

Fast tracking PEP-seeking MSM into PrEP programs is also supported by the fact that HIV events occurred several years after PEP consultation, which suggested either sustained or recurrent risk over time. In both cases, PrEP use and—more importantly—users participation in PrEP programs could be an effective approach to provide tailored prevention measures to MSM presenting varying risk patterns over time (e.g., daily versus intermittent PrEP). Institutional efforts should be thus targeted at linking and retaining PEP-seeking MSM in prevention programs offering PrEP counseling, STI screening and other services related to sexual health care. To that end, approaches similar to those found effective for MSM engagement in HIV care—such as the identification of vulnerable populations, knowledge of particular sexual practices (e.g., chemsex), respectful attitude from healthcare providers, or well-defined strategies for linkage to care—should be considered[23].

This study investigated the long-term risk of HIV infection in individuals seeking PEP. In contrast to most previously published studies[4–12], our study was larger and included a diverse PEP seekers population (heterogeneous transmission groups, i.e., inclusion of both MSM and heterosexuals), thereby ensuring wider applicability. We used data linkage to mitigate loss to follow-up bias and were able to collect outcome data that occurred—in some cases—more than 10 years after PEP consultation.

This study, however, has some limitations. A first issue is related to the retrospective design, which may lead to outcome misclassification (i.e., seropositive individuals misclassified as HIV-negative). To minimize this bias, however, we linked data to external cohorts showing high participation rates (SHCS: 84%)[24], and high representativeness of the overall HIV-positive population in Switzerland (including 75% of individuals living with HIV on antiretroviral therapy)[25,26]. This approach enabled us to nearly double the sample of HIV events identified by internal linkage: of 22 included links, 10 were retrieved exclusively through external linkage. Second, because the PEP database was not designed as a cohort study (i.e., no systematic data collection), some endpoints (such as possible PEP failures) were difficult to assess. Access to data from two large ongoing cohort studies (SHCS and ZPHI), however, enabled us to obtain a much more granular picture on these endpoints and contributed to improve data quality. Third, data collection for the PEP database stopped end of 2013, which may have limited the representativeness of the PEP-seeker population presented in this study, since recent trends—such as the emergence of chemsex or the use of PrEP – could not be fully captured by our analysis. With respect to the latter, however, widespread PrEP use in Switzerland started relatively late (i.e., around 2018) compared to other countries, as supported by an online survey performed in 2017 reporting PrEP use in only 82/1893 (4.3%) of the participants[27]. Thus, the lack of information on PrEP use may only be relevant for a small fraction of events that occurred in 2018 and later (4 out of 22 HIV events). Results generalizability may have also been hampered by the fact that PEP seeking outside the USZ was difficult to assess. Although there are no surveillance data on the total of PEP prescriptions at the regional level, empirical evidence and several surveys suggest that the USZ was one of the main PEP providers in the Zurich area during this time period. Most notably, a series of longitudinal

surveys among MSM found that between 2007–2014, PEP uptake ranged between 2.6 and 8% (mean: 5%)[28]. According to a previous report combining several data sources to estimate the size of local MSM populations[20], the number of MSM in the Zurich area was 16000 (95% credible interval: 14,300–16,500). Combining results on PEP uptake with local MSM population estimates corresponded to an estimated number of 416 to 1280 (mean: 800) PEP prescriptions occurring in Zurich within that timeframe. This suggests that the 348 MSM included in our dataset represent a substantial proportion of the total MSM population seeking PEP during the observation time. Additionally, in-depth analysis of the SHCS and ZPHI (i.e., SHCS: data on previous antiretroviral therapy [either PEP or antiretroviral therapy for HIV]; ZPHI: data on previous PEP), however, did not identify PEP consultations other than those retrieved by our linkage. Although this is only valid for PEP seekers with HIV events, it may nonetheless indicate that the USZ was one of the main PEP providers between 2007 and 2013. Fourth, because the HIV epidemic in Switzerland is mainly driven by MSM[29], and because heterosexuals tend to be underrepresented in the SHCS[25,30], outcome ascertainment bias may have occurred to some extent. However, this risk of bias was deemed low, since heterosexuals reluctant to participate to the SHCS would also most likely decline data collection for the PEP-USZ database, thereby leading to an underrepresentation on both sides of the linkage. Fifth, when assessing the long-term risk of HIV, we did not take into account potential differences in follow-up time. This limitation, however, had no impact on the two main findings of this paper, i.e., that all HIV infections occurred in MSM (who presented a follow-up times distribution similar to non-MSM PEP seekers—Supplementary Fig. 1) and that a considerable fraction of infection events occurred several years after PEP consultation. A time-to-event analysis was conducted to explore HIV rates in some specific subgroups (Supplementary Table 1). These findings, however, need to be interpreted with caution, since the lack of systematic follow-up for outcome ascertainment may have led to an underestimation of the number of HIV events, and to an overestimation of the number of participants considered at risk at the time of analysis. Hence the provided incidence rate estimates are best interpreted as lower bounds for the real incidence in these populations. Finally, our linkage method only used a few discriminative variables, which may increase the risk of false-negative and false-positive links (since any mismatch due to data error would lead to discard a true or keep a false potential pair, respectively). To address this, all 34 potential pairs retrieved through external linkage were further assessed, thereby leading to the exclusion of 9 false-positive links. Additionally, a simulation of our linkage using two pre-existing registries estimated the proportion of false-negative at 5.4%. Overall, the weaknesses of this study are thus counterbalanced by its key strength: most notably, the linkage between 3 databases provided detailed data on multiple transmission groups and long follow-up times not available in other studies.

In this retrospective data linkage study, the proportion of PEP seekers tested positive for HIV after PEP intake was 2.3%, and this proportion reached 6.3% in MSM. This study identified that most seroconversions occurred 4 years after PEP consultation, thereby bringing long-term insights into the risk of contracting HIV following PEP seeking. Those who seroconverted were all MSM, which should strongly encourage early consideration of PrEP in MSM after a first episode of PEP.

## Methods

We followed the RECORD guidelines for the REporting of studies Conducted using Observational Routinely collected health Data[31]. This data linkage study was approved by the local ethical committee (canton of Zurich, Switzerland—Registration number: 2019-00033), as were the PEP database study (registration number:

2013-0006) and the ongoing cohorts used for data linkage (Swiss HIV Cohort Study (SHCS): registration number EK-793; Zurich Primary HIV Infection (ZPHI) study: registration number: EK-1452).

**Study design and setting.** This is a retrospective study involving data linkage between 3 pre-existing databases, i.e., the PEP-USZ database, the ZPHI and the SHCS. The PEP-USZ database has been described elsewhere[32]: in brief, it consists of data from 975 individuals who sought non-occupational PEP prescription at the USZ between 2007 and 2013 (corresponding to 1051 consultations). For each PEP consultation occurring within this timeframe, data routinely collected by the attending physician were extracted from the USZ electronic patient record system and entered in the PEP-USZ database (Supplementary Note 1). Data collection occurred retrospectively (i.e., after 2013). The ZPHI is a prospective, monocentric cohort study established in 2002 at the USZ, which follows longitudinally individuals aged ≥18 years with a documented primary HIV infection (i.e., acute or recent HIV infection)[33,34]. As of July 2020, more than 450 patients were enrolled. Finally, the SHCS is a prospective, multicentric cohort study enrolling HIV-infected individuals aged ≥18 years in Switzerland since 1988[26]. As of July 2020, the SHCS included nearly 21,000 individuals, of which 9735 were still being actively followed. The region of Zurich accounts for 7701 SHCS participants, with follow-up data collected either at the USZ or in the private practice setting.

The PEP-USZ database was considered the main database, i.e., the 975 records included in this database constituted the main study cohort. The other databases were only used to obtain follow-up data through data linkage, i.e., to assess the occurrence of HIV infection in PEP seekers.

Participants involved in this project all provided consent, either for the use of their study data (ZPHI and SHCS cohort) or of their routinely collected clinical data (PEP-USZ database). More specifically, the SHCS consent form describes the possible further use of health-related data for research purposes and the linking of participant data to other sources of health-related data. For the PEP-USZ database, patients were asked at hospital presentation for PEP consultation to sign a form describing the possible further use of their health-related data for research purposes.

Data constituting the PEP-USZ database were collected using Microsoft Access; Data constituting the Zurich Primary HIV study (ZPHI) and Swiss HIV Cohort Study (SHCS) databases were collected using Microsoft Access and Oracle, respectively.

**Study participants.** All HIV-negative individuals who sought non-occupational PEP prescription between 2007 and 2013 at the USZ were considered eligible. Individuals seeking PEP for non-consensual sex or those with a positive HIV test at hospital presentation were not considered. No restriction regarding the adequacy of PEP prescription or subsequent PEP intake were applied.

**Study variables.** The primary outcome was defined as the occurrence of a documented HIV infection, using the following pre-specified criteria: (1) HIV infection diagnosed at the USZ with subsequent enrollment in the ZPHI or USZ-specific part of the SHCS (determined by internal data linkage) and/or (2) enrollment in the non-USZ part of the SHCS (i.e., private practice setting in Zurich or any other SHCS center in Switzerland, determined by external data linkage). For each HIV infection, we retrieved date(s) of PEP consultation(s) and date of HIV diagnosis.

We were also interested in establishing in each case a possible infection window, which represented the time period between the earliest possible infection date and HIV diagnosis. The earliest possible infection date was defined as the date of the last documented (or self-reported) negative HIV test. In individuals with a documented acute or recent infection (i.e., in those enrolled in the ZPHI), we used the estimated time of infection instead of the last HIV-negative test. The estimated time of infection was calculated by means of clinical and laboratory criteria, as reported previously[35]. We also assessed the occurrence of possible PEP failures, which were defined as events with an infection window starting within 3 months of PEP consultation[36].

To characterize subgroups with a higher long-term risk of HIV infection, the following variables were retrieved: age, sex, nationality, group at higher risk of HIV transmission (MSM), whether PEP was indicated at the time of PEP consultation (according to local prescription algorithm), and repeated PEP intake. The latter was defined as self-reported repetitive PEP seeking (elicited from medical history) and/or identification of multiple PEP seeking visits during study period (2007-2013).

Adherence to PEP (i.e., prescribed but not taken, prescribed but inadequately taken) was used to assess the plausibility of transmission events. All risk factors and confounders were predefined and based on data availability from the pre-existing PEP-USZ database[32]. Only factors related to individuals seeking PEP (as opposed to the episodes triggering PEP consultation) were considered.

**Data sources and linkage methods.** As both the USZ-specific part of the SHCS and the ZPHI share a common unique identifier with the PEP-USZ database, this institutional linkage was defined as "internal linkage". Linkage with the non-USZ part of the SHCS (i.e., private practice setting in Zurich or any other SHCS center in Switzerland), however, was defined as "external" and used the P3RL method (Fig. 4)[15].

This approach used a third party to perform probabilistic data linkage after predefined variables were pre-processed and encrypted by the responsible centers (i.e., USZ research team for the PEP-USZ database and the SHCS datacenter for the non-USZ SHCS database). Record linkage was undertaken using an encrypted, non-unique, personal identifying information (i.e., complete date of birth) and the following plain linkage variables: (1) gender, (2) an anonymous code used in the SHCS cohort (i.e., initial of the patient first name combined with the number of characters of the first name, e.g., Alan = A4), and (3) only the initial of the patient first name. The coherence of specific time variables was also considered during linkage, i.e., HIV diagnosis could not occur before PEP consultation. Date of birth encryption used a procedure based on Keyed-Hash Message Authentication Code (HMAC), which handles common systematic errors such as swapping days with months or number transpositions. Probabilistic record linkage was performed by assigning weights to potential pairs fulfilling a predefined set of rules (i.e., matching encrypted date of birth, gender, initial of first name, anonymous code, and time plausibility). Three weight thresholds were defined to categorize potential pairs: those with a total weight <70 were rejected; pairs with total weight between 70 and 89 were considered as "questionable" links, those between 90 and 99 as "possible" and those ≥100 as "definite"[37]. Linkage probability was thus not only affected by total weight, but also by the number of alternative links (i.e., links with match on date of birth and gender only). Because the linkage procedure was based on only one highly discriminative variable (date of birth), we contacted individual SHCS centers to perform additional data checks for all non-rejected pairs. Finally, to estimate the expected false-negative proportion yielded by the linkage method, we performed a simulation study using two pre-existing registries (HIV data consisting

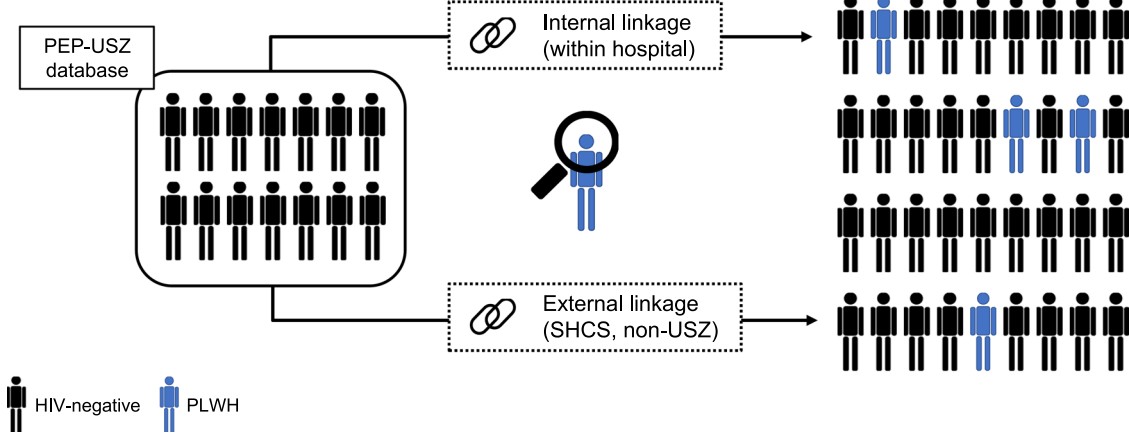

**Fig. 4 Study design.** Records retrieved from the PEP-USZ database were linked with 2 cohort studies, using a common unique identifier (internal linkage: ZPHI and USZ-specific part of SHCS) or a privacy preserving probabilistic linkage (external linkage: SHCS, non-USZ part), with the aim to identify PLWH (blue person). PEP: post-exposure prophylaxis; PLWH: person living with HIV; SHCS: Swiss HIV Cohort Study; USZ: University Hospital of Zurich; ZPHI: Zurich Primary HIV Infection study.

of 2,868 records, and Cancer Registry data consisting of 90,276 records). The simulation linkage between these registries used the exact same variables as those used in our study. The proportion of false negatives was estimated by comparing the results of the simulation linkage study with a gold-standard, i.e., with the results from a previous linkage performed between these registries using several highly discriminative variables (such as date of birth, date of death, names, or place of residence)[15].

**Statistical methods**. We did not perform sample size calculations, as the size of the PEP seekers cohort was defined by the fixed size of the available PEP-USZ database (data from 975 individuals collected over 6 years).

The primary analysis was descriptive, i.e., we assessed the long-term occurrence of HIV infection in USZ PEP seekers using data linkage. Categorical variables were expressed as proportions, continuous as median and interquartile range. For the assessment of HIV events with regard to the overall population or to specific subgroups, proportions with their 95% confidence intervals (CI) were computed.

Depending on the number of outcome events, we planned to conduct univariable and multivariable logistic regression to explore which factors were associated with long-term risk of HIV diagnosis. Finally, we assessed the validity of the PEP prescription algorithm, i.e., whether the algorithm used at the time of PEP consultation was able to identify individuals presenting a long-term risk of HIV infection. To achieve this, we determined the occurrence of HIV infection in those for whom PEP was indicated and explored the association between PEP indication and HIV infection using logistic regression. All statistical analyses were conducted in R, version 3.6.1. Two-sided tests were performed, and a level of significance of 0.05 was used.

**Reporting summary**. Further information on research design is available in the Nature Research Reporting Summary linked to this article.

## Data availability
Datasets analyzed during the current study and used to generate Table 1, Figs. 2 and 3, and supplementary information are not publicly available due to the sensitive nature of the data yielded by this small, highly representative, individual-level dataset (see also: http://www.shcs.ch/294-open-data-statement-shcs). Source data are thus not provided with this paper. Investigators with a request for selected data should send a proposal to the Swiss HIV Cohort Study (SHCS) address (www.shcs.ch/contact). The provision of data will be considered by the Scientific Board of the SHCS and the relevant study team. Data provision is subject to Swiss legal and ethical regulations, and will be detailed in a material and data transfer agreement.

## Code availability
Study protocol and code can be made available from the corresponding author (FH) on reasonable request.

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

## Acknowledgements
This study was financed within the framework of the Swiss HIV Cohort Study (Swiss National Science Foundation grant #177499). The study was supported by the SHCS in the context of the SHCS project 845. R.D.K. and F.H. were supported by the Swiss National Science Foundation (Grant #BSSGI0_155851). The funding organizations had no role in the design and conduct of the study; in the collection, management, analysis, or interpretation of the data; or in the preparation, review, or approval of the manuscript. Data were collected by five Swiss University Hospitals, two Cantonal Hospitals, 15 affiliated hospitals and 36 private physicians (listed in http://www.shcs.ch/180-health-care-providers). We thank the patients who participated in the SHCS and the ZPHI; the physicians and study nurses for excellent patient care; A. Scherrer, A. Traytel, S. Wild and K. Kusejko from the SHCS Data Center for data management; and D. Perraudin and M. Amstad for administrative assistance.

## Author contributions
F.H., H.F.G., S.K.R., J.S.F. and R.D.K. participated in study conception and design, data acquisition and interpretation, and critical revision of the manuscript. F.H. drafted the first manuscript. F.H. and R.D.K. performed the statistical analyses. A.Sc., A.Sp. and K.S. performed the external linkage. H.F.G., C.H., A.Co., E.B., A.Ca., M.C., M.S., A.M., H.H. and D.L.B. participated in data acquisition and critical revision of the manuscript. J.R. and S.B. participated in data analysis, interpretation, and critical revision of the manuscript. F.H. obtained funding. R.D.K. supervised the study. All authors listed on the title page have read the manuscript, attest to the validity and legitimacy of the data and its interpretation, and agree to its submission.

## Competing interests
H.F.G. has served as a consultant or advisor to Merck & Co, Inc, ViiV Healthcare, and Gilead Sciences, Inc, as a member on data and safety monitoring boards for Merck & Co, Inc. and has received research grants from Gilead Sciences, Inc. payed to his institution. The institution of EB received fees for his participation to advisory boards and travel grants from Gilead Sciences, MSD, ViiV Healthcare, Abbvie, Pfizer, and Sandoz. D.L.B. received honoraria and travel grants unrelated to the submitted work from Merck & Co, Gilead Sciences, and ViiV Healthcare. J.S.F. received research grants unrelated to the submitted work from Merck & Co, Gilead Sciences, and ViiV Healthcare. The remaining authors declare no competing interests.

## Additional information

## the Swiss HIV Cohort Study
K. Aebi-Popp[4], A. Anagnostopoulos[1], M. Battegay[15], E. Bernasconi[6], J. Böni[3], D. L. Braun[1], H. C. Bucher[16], A. Calmy[7], M. Cavassini[8], A. Ciuffi[17], G. Dollenmaier[18], M. Egger[13], L. Elzi[19], J. Fehr[1,2], J. Fellay[20,21], H. Furrer[4], C. A. Fux[5], H. F. Günthard[1], D. Haerry[22], B. Hasse[1], H. H. Hirsch[23], M. Hoffmann[9], I. Hösli[24], M. Huber[3], C. R. Kahlert[9,25], L. Kaiser[7], O. Keiser[13], T. Klimkait[23], R. D. Kouyos[1,3], H. Kovari[1], B. Ledergerber[1], G. Martinetti[26], B. Martinez de Tejada[27], C. Marzolini[15], K. J. Metzner[1,3], N. Müller[1], D. Nicca[2], P. Paioni[28], G. Pantaleo[19], M. Perreau[19], A. Rauch[4], C. Rudin[29], A. U. Scherrer[1], P. Schmid[9], R. Speck[1], M. Stöckle[15], P. Tarr[30], A. Trkola[3], P. Vernazza[9], G. Wandeler[4], R. Weber[1] & S. Yerly[7]

[15]Department of Infectious Diseases and Hospital Epidemiology, University Hospital Basel, University of Basel, Basel, Switzerland. [16]Basel Institute for Clinical Epidemiology and Biostatistics, University Hospital Basel, University of Basel, Basel, Switzerland. [17]Institute of Microbiology, University Hospital Lausanne, University of Lausanne, Lausanne, Switzerland. [18]Centre for Laboratory Medicine, Canton St. Gallen, St. Gallen, Switzerland. [19]Division of Immunology and Allergy, Centre Hospitalier Universitaire Vaudois, University of Lausanne, Lausanne, Switzerland. [20]School of Life Sciences, EPFL, Lausanne, Switzerland. [21]Precision Medicine Unit, Lausanne University Hospital and University of Lausanne, Lausanne, Switzerland. [22]Positive Council, Zurich, Switzerland. [23]Department Biomedicine - Petersplatz, University of Basel, Basel, Switzerland. [24]Clinic for Obstetrics, University Hospital Basel, University of Basel, Basel, Switzerland. [25]Children's Hospital of Eastern Switzerland, St. Gallen, Switzerland. [26]Cantonal Institute of Microbiology, Bellinzona, Switzerland. [27]Department of Obstetrics and Gynecology, University Hospital Geneva, University of Geneva, Geneva, Switzerland. [28]University Children's Hospital, University of Zurich, Zurich, Switzerland. [29]University Children's Hospital, University of Basel, Basel, Switzerland. [30]Kantonsspital Baselland, University of Basel, Basel, Switzerland.

