## [Peer Review File · Nature Communications]

REVIEWER COMMENTS

Reviewer #1 (Remarks to the Author):

With declines in HIV incidence in many settings identifying people who are eligible for still at considerable risk of HIV for provision of effective HIV pre-exposure prophylaxis is key. Having had one substantial episode of risk would be a good way to identify this group. Taking HIV post exposure prophylaxis as such a marker. Data from STI surveillance in the UK has identified taking PEP as a risk for HIV acquisition in MSM and is one of the criteria for offering PrEP. To date most information on non-occupational PEP as a marker of risk has come from sexual health clinics and does not include a large population of heterosexuals. It is therefore good to see such a robust study looking at a wider population. Very nice description of the data linkage and defining the outcome and timeframe of new HIV infections.

My main concerns are around generalisability, ethics, and some seemingly post-hoc analysis. I suggest that the reporting follow STROBE guidelines

https://www.strobe-statement.org/fileadmin/Strobe/uploads/checklists/STROBE_checklist_v4_cohort.pdf

This is an important paper, however would be strengthened with the authors revising the manuscript to clarify some of the following issues:

Methods

Population: I have a few questions for the authors on the population to help unpack the generalisability of these findings

1. Where is non occupational PEP usually prescribed? What proportion of all PEP users in that time period would the 975 PEP-USZ be? Where else was PrEP sought?
2. What were the eligibility criteria for all the Swiss HIV cohort and the primary infection cohort? and what is the response rate?

Analysis:

1. Was it possible to ascertain any PrEP use in those at highest risk?
2. How were confounders decided on? What was the pre-specified analysis

Ethics:

1. Where is the ethics statement for this secondary data analysis. Do the authors have ethics?
2. Why wasn't PrEP offered to the PEP-USZ participants when its effectiveness was proven

Results

1. In keeping with the STROBE recommendations for reporting cohorts I would like to see the number of individuals at each stage – how many were eligible for PEP-USZ, how many consented and how many contribute data to this analysis –
2. Table 1 compares HIV + and HIV – and it would be better to have some idea of the comparison between included and non-included participants
3. The in-depth analysis of the source of infection seems fairly irrelevant and I am not sure why this analysis was done or what it adds with an n=22. This looks like a post-hoc analysis, there is no comparison with the 953 PEP users who didn't get HIV – – and the study was not designed to answer this question - I would strongly suggest removing it –

Discussion

1. This should not repeat the results – it would be better to highlight that in this longer and more

diverse PEP seeking cohort the key/ salient findings were – e.g. HIV incidence occurred much later than is usually expected, confirming that incident HIV occurred almost exclusively in MSM PEP users and that clinical assessment of eligibility for PrEP and multiple PrEP use were predictors of incidence suggesting that this group should be fast tracked into PEP into PrEP programmes

2. I would suggest removing the discussion of the in-depth analysis. The study was not designed to answer this question – it's not clear how the those who had contracted HIV compared with those that didn't etc

3. It would be useful to comment on how these finding would or wouldn't impact on current European and swiss PrEP guidance? Is PrEP being offered to PEP users with high risk sexual behaviours?

The limitations would be beneficial to reflect on

1. How representative the PEP cohort was (see my points above)

2. Are there any potential biases in the outcome ascertainment. Are MSM more likely to be diagnosed with HIV (what is the late diagnosis rate in heterosexuals or IDUs in Switzerland)? Are MSM more likely to have been linked? Are the other groups more mobile? Less likely to be in the swiss cohort? More likely to die before testing for HIV?

Abstract

1. Can the abstract follow STROBE reporting guidelines?

Reviewer #2 (Remarks to the Author):

This paper describes a data linkage study to estimate the long-term HIV risk among individuals seeking PEP from a tertiary setting in Switzerland. The authors find that among 975 individuals who sought PEP, 22 had evidence of HIV diagnosis from a linked cohort study, with the proportion highest among MSM (6.3%). This study provides important evidence which supports the transition from PEP to PrEP among MSM. The paper is well written and the methodology is sound.

Introduction

Page 5 Line 9. Is there only one recommended regimen / formulation for PEP in Switzerland. Can the authors provide a reference to the relevant PEP prescribing recommendations / guidelines?

Page 6 Line 9. "We assumed that HIV infections would be clustered shortly after the time of PEP consultation." It is not clear, at this point in the manuscript, what the relevance of this assumption is for the analysis.

Methods

Page 7 Lines 10-14. Can you also describe what data is routinely collected during a PEP consultation for the PEP-USZ database?

Page 7 Line 24. Did participants explicitly consent to data being linked across different studies?

Page 8 Line 4. By 'hospital admission' do you mean at time of admission for their PEP consultation?

Page 10 Line 16. The authors mention the possibility of repeated PEP consultations, with both consultations recorded in the PEP-USZ database. Does that mean that the 975 included PEP records refer to first-time PEP consultations only?

Page 10 Line 22. The authors state "We assessed whether the decision to prescribe PEP was appropriate. i.e. whether the algorithm used at the time of PEP consultation was able to correctly identify individuals presenting a long-term risk of HIV infection." I am not sure whether prescribing PEP is only *appropriate* if there exists long-term risk of HIV infection. The language used by the authors implies it was only appropriate to prescribe PEP among those who eventually seroconverted. More likely, the authors sought to assess the validity or appropriateness of the algorithm (not the decision to prescribe PEP in any one instance).

Results

Page 13 Line 4: "HIV infections occurred in 2.3%..." I would change to "Evidence of HIV infection..", as you do not have HIV-negative outcomes for the rest of the cohort (some may have been diagnosed and not enrolled in any of the linked cohorts).

Page 13 Line 6. As above, I would refrain from calling those whom you do not have linked HIV diagnosis data for as "HIV-negative", as this is not confirmed.

Table 1. Please specify when age was recorded, i.e. at initial PEP consultation, or at last PEP consultation date? The latter is used to estimate median time between last PEP date and HIV diagnosis date.

Page 16 Line 20. As above, in your analysis of the indication for PEP algorithm, I would reconsider describing the decision to prescribe PEP as 'appropriate' or not.

Discussion

Page 17 Line 8. 3.8 represents an odds ratio, so it is not correct to say a 3.8-fold increase in the *risk of HIV infection*.

Page 17 Lines 4-6. The authors describe that HIV diagnoses mostly occurred several years after PEP consultation, and then raise the point that "MSM should be encouraged to transition to PrEP as early as after a first PEP episode", which seems somewhat conflicting. Perhaps the authors could elaborate on their rationale for immediate PrEP transition, given the long-term risk. Some further discussion around transition between periods of HIV risk would be welcomed.

Given the main findings and recommendations of this paper seem to be in support of transitioning PEP users to PrEP, some discussion around how those seeking PEP can be effectively and quickly linked into PrEP programs, and be supported to remain on PrEP for long periods of time, would strengthen the discussion section.

Are the authors able to estimate an HIV incidence rate following PEP consultation, i.e. calculate person time among all PEP seekers, assuming those with no evidence of HIV diagnosis via one of the cohort studies remained HIV-negative until October 2019. This would help the reader contextualise the risk of long-term HIV among PEP seekers relative to all MSM.

Reviewer #3 (Remarks to the Author):

This paper examines the risk of HIV in a population based study of PEP users. While the efficacy of PEP in the short -term has been well established, long term effects and behaviour have not been well elucidated. The assessment of ongoing HIV risk is particularly pertinent given ongoing risk and the advent of effective prevention using PrEP. This clear and concise paper addresses these issues using appropriate and valid methodologies.

The data utilised in this study are derived from comprehensive PEP and HIV data sources. However, other than shorter duration of follow-up, it is not clear why USZ data are limited to only 2013. Is the 2013 end date to define the cohort or also data censoring (see comment below re Figure 3)? Within the Swiss context declining HIV incidence in MSM has been documented between 2008-2013 (<https://doi.org/10.1371/journal.pone.0131828>), and it is likely these trends have continued. It would be interesting to see if this is reflected in decreasing HIV in MSM accessing PEP, noting that event numbers are very small. Additionally, including a comparison of HIV risk in this population to that in Swiss MSM, would help contextualise the conclusion that MSM using PEP group should be specifically targeted for PrEP.

The data linkage methodology used in this study is the most relevant for this type of study question. The description of linkage process is clear and sufficient. The ability to assess HIV linkages through review of clinical records and eliminate false positives is a strength of this study. The simulation methodology to assess false negatives could be expanded in the methods. It would also be beneficial to include uncertainty bounds for the assessment of HIV risk in this population. While the usual method of analysis for this type of data would be time to event, the lack of routine screening for events, along with the other reasons described by the authors, supports summary of events as risk.

Specific comments:

Pg 16, Ln 19: for clarity "thereby impairing meaningful multivariable regression"

Pg 16 Ln 24: does this include adjustment for PEP post 2013?

Pg 26, Figure 3. Were there any repeat occurrences of PEP between 2013 and 2018?

REVIEWER COMMENTS

Preliminary note: in this point-by-point reply, changes made to the manuscript are indicated below each answer to the reviewers, under a section entitled "Related revised manuscript text". When copy-pasting these revised parts of the manuscript, we removed the in-text citations, so as to avoid mixing the manuscript references with references provided in this point-by-point reply only.

Reviewer #1 (Remarks to the Author)

With declines in HIV incidence in many settings identifying people who are eligible for still at considerable risk of HIV for provision of effective HIV pre-exposure prophylaxis is key. Having had one substantial episode of risk would be a good way to identify this group. Taking HIV post exposure prophylaxis as such a marker. Data from STI surveillance in the UK has identified taking PEP as a risk for HIV acquisition in MSM and is one of the criteria for offering PrEP. To date most information on non-occupational PEP as a marker of risk has come from sexual health clinics and does not include a large population of heterosexuals. It is therefore good to see such a robust study looking at a wider population. Very nice description of the data linkage and defining the outcome and timeframe of new HIV infections.

My main concerns are around generalisability, ethics, and some seemingly post-hoc analysis. I suggest that the reporting follow STROBE guidelines: https://www.strobe-statement.org/fileadmin/Strobe/uploads/checklists/STROBE_checklist_v4_cohort.pdf

This is an important paper, however would be strengthened with the authors revising the manuscript to clarify some of the following issues:

Methods

Population: I have a few questions for the authors on the population to help unpack the generalisability of these findings

1. Where is non occupational PEP usually prescribed? What proportion of all PEP users in that time period would the 975 PEP-USZ be? Where else was PEP sought?

Reply: Thanks for raising these important points. Between 2007 and 2013 (time period constituting the PEP-database), access to PEP in the city of Zurich was mainly provided by the emergency departments of large tertiary care hospitals (in Zurich: University hospital of Zurich (USZ), Cantonal Hospital of Zurich (Triemli)) or sexual health clinics. Although there are no surveillance data on the total of PEP prescriptions at the regional level, empirical evidence and several surveys suggest that the USZ was one of the main PEP providers in the Zurich area during this time period. Most notably, a series of longitudinal surveys among MSM found that between 2007-2014, PEP uptake ranged between 2.6 and 8% (mean: 5%).¹ According to a previous report combining several data sources to estimate the size of local MSM populations,² the number of MSM in the Zurich area was 16000 (95% CrI: 14300 – 16500). Combining results on PEP uptake with local MSM population estimates corresponded to an estimated number of 416 to 1280 (mean: 800) PEP prescriptions occurring in Zurich within that timeframe. This suggests that the 348 MSM included in our dataset represent a substantial proportion of the total MSM population seeking PEP during the observation time.

In our manuscript, we acknowledge the difficulty to assess PEP episodes occurring outside the USZ (Discussion, p.18). It is moreover good to note, that both the SHCS and ZPHI collect information on PEP episodes prior to HIV infection. In-depth review of the SHCS and ZPHI data did not identify PEP consultations other than those retrieved by our linkage (p.14, end of first paragraph). Although this is

only valid for PEP-seekers with HIV events, it might further support that the USZ was one of the main PEP providers between 2007 and 2013. This information has been added to the limitation section (p.18).

RELATED REVISED MANUSCRIPT TEXT:

"Results generalizability may also have been hampered by fact that PEP seeking outside the USZ was difficult to assess. Although there are no surveillance data on the total of PEP prescriptions at the regional level, empirical evidence and several surveys suggest that the USZ was one of the main PEP providers in the Zurich area during this time period. Most notably, a series of longitudinal surveys among MSM found that between 2007-2014, PEP uptake ranged between 2.6 and 8% (mean: 5%). According to a previous report combining several data sources to estimate the size of local MSM populations, the number of MSM in the Zurich area was 16000 (95% credible interval: 14300 – 16500). Combining results on PEP uptake with local MSM population estimates corresponded to an estimated number of 416 to 1280 (mean: 800) PEP prescriptions occurring in Zurich within that timeframe. This suggests that the 348 MSM included in our dataset represent a substantial proportion of the total MSM population seeking PEP during the observation time. Additionally, in-depth analysis of the SHCS and ZPHI (i.e. SHCS: data on previous antiretroviral therapy (either PEP or antiretroviral therapy for HIV); ZPHI: data on previous PEP), however, did not identify PEP consultations other than those retrieved by our linkage. Although this is only valid for PEP-seekers with HIV events, it may nonetheless indicate that the USZ was one of the main PEP providers between 2007 and 2013."

MANUSCRIPT LOCATION: p.18, lines 2-17

2. What were the eligibility criteria for all the swiss HIV cohort and the primary infection cohort? and what is the response rate?

Reply: the eligibility criteria of the ZPHI and SHCS are described on page 6 (Methods section, study design and setting). The ZPHI enrolls individuals aged ≥ 18 years with a documented primary HIV infection (i.e. acute or recent HIV-infection). The SHCS is targeted at HIV-infected individuals aged ≥ 18 years. ZPHI participation rates have been estimated around 90-95% (unpublished data). SHCS participation/response rates (reported to reach 84% in a report from 2012)³ are briefly mentioned in the limitation section (p.17), but without further details. Thus we added this information to the manuscript.

RELATED REVISED MANUSCRIPT TEXT:

"To minimize this bias, however, we linked data to external cohorts showing high participation rates (ZPHI: 90-95% (unpublished data), SHCS: 84%), and high representativeness of the overall HIV-positive population in Switzerland (including 75% of individuals living with HIV on antiretroviral therapy)."

MANUSCRIPT LOCATION: p.17, lines 13-15

Analysis:

1. Was it possible to ascertain any PrEP use in those at highest risk?

Reply: This is a very good point. We fully agree that data on prior PrEP use would be of added value in our analysis, e.g. as a marker of high risk behaviour around the time of infection. This information is unfortunately neither available from the SHCS, nor the ZPHI, nor the PEP-database. It is important to keep in mind, however, that PrEP use in Switzerland started rather late compared to other countries, such as the UK or the USA: the median PrEP start date (prior to study enrollment) among 1130 participants of the SwissPrEPared cohort study was November 2018 (IQR: March 2018 –

June 2019; unpublished data from the SwissPrEPared cohort study, NCT03893188). This rather late PrEP uptake in Switzerland was also reported in an online survey performed in Jan 2017 among Swiss Grindr® users: only 82/1893 (4.3%) were taking PrEP at the time of the survey.⁴ Thus, the lack of information on PrEP use may only be relevant for a small fraction of events that occurred in 2018 and later (4 out of 22 HIV events) (individuals number 4, 10, 13, 16 – Figure 3) and of no relevance in the PEP-database (since it was limited to 2013).

RELATED REVISED MANUSCRIPT TEXT :

"Third, data collection for the PEP database stopped end of 2013, which may have limited the representativeness of the PEP-seeker population presented in this study, since recent trends – such as the emergence of chemsex or the use of PrEP – could not be fully captured by our analysis. With respect to the latter, however, widespread PrEP use in Switzerland started relatively late (i.e. around 2018) compared to other countries, as supported by an online survey performed in 2017 reporting PrEP use in only 82/1893 (4.3%) of the participants.³² Thus, the lack of information on PrEP use may only be relevant for a small fraction of events that occurred in 2018 and later (4 out of 22 HIV events)."

MANUSCRIPT LOCATION: p.17-18, lines 21-26; and 1-2.

2. How were confounders decided on? What was the pre-specified analysis

Reply: All risk factors and confounders were pre-defined in our study protocol and based on data availability from a pre-existing PEP-database (published in 2017 by Marzel et al.).⁵ Since we expected outcome events to be scarce, we planned to limit the regression analysis to a few independent variables (rule of the thumb would be 1 covariate per 10 outcome events, i.e. in our case max 3 covariates). We eventually retained "MSM" and "repeated PEP seeking", not only because previous literature identified these variables as significant risk factors, but also because these were the only risk factors reported by Marzel et al. that related to the individuals themselves (and not to the exposure/episode triggering PEP consultation). As for the assessment of "PEP indication", it was a pre-specified risk factor in our study protocol and thus automatically retained in the analysis. "PEP adherence" was identified as a potential confounder, but adherence data were not systematically collected in the PEP database (information available for only 36% of all consultations), so that this variable could not be assessed.

We agree that this is important to clarify these aspects and adapted the manuscript accordingly.

RELATED REVISED MANUSCRIPT TEXT:

"All risk factors and confounders were predefined and based on data availability from the pre-existing PEP-USZ database. Only factors related to individuals seeking PEP (as opposed to the episodes triggering PEP consultation) were considered."

MANUSCRIPT LOCATION: p.8, lines 10-12

Ethics:

1. Where is the ethics statement for this secondary data analysis. Do the authors have ethics?

Reply: this is a very important point and we are sorry to notice that the wording in our manuscript was not clear enough. All databases/cohorts used in this linkage project have their own ethical approval and we also obtained ethical approval for the linkage project itself (as mentioned p.6: "This data linkage study (i.e. the linkage of 3 different databases) was approved by the local ethical committee

(canton of Zurich, Switzerland – Registration number: 2019-00033), as were the ongoing cohorts used for data linkage"). Regarding participant consent, there is a mention further in the text on p.7: "Participants involved in this project all provided consent, either for the use of their study data (ZPHI and SHCS cohort) or of their routinely collected clinical data (PEP-USZ database)." As it is often the case for retrospective studies based on routinely collected data, individuals included in our analysis were not individually contacted to obtain consent for the linkage of their health-related data, because this would not be feasible, and because further data use for scientific purposes was already covered by the informed consent forms signed at the time of study entry (e.g. SHCS) or hospital presentation (e.g. PEP database).

Here some additional details:

1. The PEP-USZ database project was approved in 2013 by the local ethical committee (canton of Zurich, Switzerland – Registration number: 2013-0006). In this retrospective analysis, data routinely collected by the physician during PEP consultation were extracted from the USZ electronic patient record system and entered in the PEP-USZ database. At hospital admission for PEP consultation, patients were asked to sign a document mentioning explicitly the possible further use of their health-related data for research purposes. The document gave the possibility to express their veto. The document was then consigned in the electronic personal record. In the PEP-USZ database, only patients who agreed on data collection were included. None of the retrieved records were excluded because of non-consent.

2. The ZPHI study was approved by the local ethical committee (canton of Zurich, Switzerland – Registration number: EK-1452). In October 2015, an amendment to carry on with prospective data collection was submitted and approved by the ethical committee. Data source for the ZPHI is the electronic patient chart system of the USZ, i.e. data collected during a regular appointment with the infectious disease specialist. Of note: all participants of the ZPHI retrieved in our analysis were SHCS participants.

3. The SHCS was approved by the local ethical committee of all participating centers including the canton of Zurich, Switzerland – Registration number: EK-793 – see also: <http://www.shcs.ch/206-ethic-committee-approval-and-informed-consent>). The consent form of the SHCS mentions explicitly the possible further use of health-related data for research purposes and the linking of SHCS participant data to other sources of health-related data.

Because this point merits clarification, we adapted the ethical statement accordingly, which now reads:

RELATED REVISED MANUSCRIPT TEXT:

"This data linkage study was approved by the local ethical committee (canton of Zurich, Switzerland – Registration number: 2019-00033), as were the PEP database study (registration number: 2013-0006) and the ongoing cohorts used for data linkage (Swiss HIV Cohort Study (SHCS): registration number EK-793; Zurich Primary HIV Infection (ZPHI) study: registration number: EK-1452)."

MANUSCRIPT LOCATION: p.6, lines 3-7

2. Why wasn't PrEP offered to the PEP-USZ participants when it's effectiveness was proven

Reply: this is a valid question, which has been partly answered above ("Analysis", first question). There are 2 important aspects that should be mentioned here:

1. The PEP-USZ database includes data on individuals seeking PEP between 2007 and 2013. Data collection was limited to 2013 due to departmental restructuring, which limited further data availability.

2. Even though PrEP approval started around 2013 in other countries, Switzerland was much slower in the scale-up of PrEP programs (see also our reply above). We cannot exclude that some of the PEP-USZ participants may have been offered PrEP after 2013 (e.g. in the case of repetitive PEP seeking) – this information, however, cannot be captured by this linkage study.

RELATED REVISED MANUSCRIPT TEXT (or Table/Figure): none

MANUSCRIPT LOCATION: none

Results

1. In keeping with the STROBE recommendations for reporting cohorts I would like to see the number of individual at each stage – how many were eligible for PEP-USZ, how many consented and how many contribute data to this analysis –

Reply: We thank the reviewer for this valuable comment. We fully agree that a flow chart detailing these figures is mandatory for cohort studies. However, there seems to be some confusion about the design of the PEP-USZ database, which definitively merits further clarification. As described on p.6 of the manuscript, the PEP-USZ database consists of routinely collected data from individuals who sought PEP between 2007 and 2013. In contrast to ZPHI and SHCS (where the data are collected prospectively), the PEP-database is based on a retrospective design, i.e. data from PEP consultations occurring between 2007-2013 were entered in a database after obtaining ethical approval in 2013. The retrospective nature of data collection is also mentioned in the limitation section (p.17, line 11). Thus, key design features of cohort studies (such as participation or retention rates) cannot be retrieved for the PEP-USZ database. Nonetheless, as a proxy for these numbers, it is worth to mention that none of the potentially eligible PEP-seekers records were excluded because of a veto/non-consent for data collection. Thus, of 975 screened patient records, all consented and participated to the analysis.

Of note: this is also the main reason why we followed the RECORD guidelines (for the REporting of studies Conducted using Observational Routinely-collected health Data – p.6 of the manuscript) and not the STROBE guidelines.

We changed the wording in the manuscript to reflect the retrospective nature of data collection in the PEP-USZ database.

RELATED REVISED MANUSCRIPT TEXT (or Table/Figure):

"Data collection occurred retrospectively (i.e. after 2013)."

MANUSCRIPT LOCATION: p.6, line 16.

2. table 1 compares HIV + and HIV – and It would be better to have some idea of the comparison between included and non-included participants

Reply: That would be of major interest, would the PEP-USZ database have been conceived as a cohort study: one would, for instance, expect a different risk profile among participants withdrawing consent or in those lost to follow-up. As mentioned earlier, in our study, data collection was retrospective and all potentially eligible records (i.e. all those who consent to data collection) were included.

RELATED REVISED MANUSCRIPT TEXT (or Table/Figure): none

MANUSCRIPT LOCATION: none

3. The in-depth analysis of the source of infection seems fairly irrelevant and I am not sure why this analysis was done or what it adds with an n=22. This looks like a post-hoc analysis, there is no comparison with the 953 PEP users who didn't get HIV – – and the study was not designed to answer this question - I would strongly suggest removing it –

Reply: We agree that this exploratory, descriptive analysis of risk factors around the time of infection was of post-hoc nature. However, our aim was not to evaluate the direct association between these factors and the occurrence of HIV events (otherwise, as rightly pointed out, a control group would be mandatory), but rather to generate some hypotheses about the factors that may – to some extent – contribute to the late occurrence of HIV events seen in our study.

That said, we agree that our manuscript may have put too much emphasis on this exploratory analysis. Following the reviewer's recommendation, this section and its related part in the discussion (see also comment below) were thus removed from the manuscript.

RELATED REVISED MANUSCRIPT TEXT (or Table/Figure):

Last part of the result section was deleted; second paragraph of discussion was substantially changed (following the reviewer's suggestion described here below, i.e. under "Discussion", comment #2).

MANUSCRIPT LOCATION: p.14 and p.15

Discussion

1. This should not repeat the results – it would be better to highlight that in this longer and more diverse PEP seeking cohort the key/ salient findings were – e.g. HIV incidence occurred much later than is usually expected, confirming that incident HIV occurred almost exclusively in MSM PEP users and that clinical assessment of eligibility for PEP and multiple PEP use were predictors of incidence suggesting that this group should be fast tracked into PrEP programmes

Reply: we thank the reviewer for pointing this out and made the changes accordingly.

RELATED REVISED MANUSCRIPT TEXT (or Table/Figure):

"In this retrospective data linkage study including a large and diverse PEP seeker population, we found that HIV infections occurred several years after PEP consultation, i.e. much later than originally assumed. Even though MSM constituted only 35.8% of PEP-seekers, infections were found exclusively in this subgroup. Clinical indicators, such as repetitive PEP seeking or indication for PEP, were associated with subsequent HIV infection".

MANUSCRIPT LOCATION: p.15, lines 2-6

2. I would suggest removing the discussion of the in-depth analysis. The study was not designed to answer this question – it's not clear how the those who had contracted HIV compared with those that didn't etc

Reply: this has been addressed above (Results, question 3). This exploratory analysis has been removed from the manuscript. As a consequence, the 2nd paragraph of the discussion has been substantially changed, as it has also been adapted to R#2, Discussion comment #2 on long term risk.

RELATED REVISED MANUSCRIPT TEXT:

"According to evidence on sexual behavior, sexual risk taking in HIV-negative MSM seems to follow distinct patterns over time: in a longitudinal analysis of data from the Multicenter AIDS Cohort Study, individuals deemed at moderate risk of seroconversion presented phases of risky behavior that lasted on average 12 months, whilst those deemed at high risk exhibited phases of 24 months. Thus, considering PEP seeking as a marker of sexual risk taking, we assumed that outcome events would be clustered 12 to 24 months after PEP seeking. Surprisingly, however, most HIV events in our study occurred several years after PEP consultation. Our study suggests thus that phases of risk may persist for longer periods than originally assumed, or that some trajectories may include recurrent phases of risk. Large cohort studies enrolling HIV-negative individuals at considerable risk of seroconversion – such as the SwissPrEPared cohort study (NCT03893188) – will bring a better understanding of sexual behavior and risk trajectories over time."

MANUSCRIPT LOCATION: p.15, lines 8-18

3. It would be useful to comment on how these findings would or wouldn't impact on current European and Swiss PrEP guidance? Is PrEP being offered to PEP users with high risk sexual behaviours?

Reply: This is a very valid point, which has already been partly addressed in the first version of our manuscript. In the latest EACS recommendations (European AIDS Clinical Society, Version 10.0, 2019), although PrEP is indicated in individuals "using PEP", there is no clear guidance as to when the transition from PEP to PrEP should occur. The Swiss guidelines (issued in 2016) recommends PrEP only in those with multiple PEP episodes. The Recommendations of the International Antiviral Society (USA Panel) do not specify any timing. We thus modified this paragraph and discussed how our study findings may impact current guidance. The revised paragraph now reads:

RELATED REVISED MANUSCRIPT TEXT:

"Although some institutions may have already started considering first-PEP-episodes in MSM as an indication for PrEP (as it is currently the case at the USZ), the exact timing of referral to PrEP programs is still lacking in most national and international recommendations. Findings from our study may thus further contribute to closing the knowledge gap regarding the timing of PEP to PrEP transition. More specifically, a first, successful PEP episode (i.e. completed without subsequent HIV infection) should trigger eligibility screening for PrEP."

MANUSCRIPT LOCATION: p.16, lines 3-9

The limitations would be beneficial to reflect on

1. How representative the PEP cohort was (see my points above)

Reply: thanks for highlighting this, the limitation section now includes a part on the representativeness of our study population (3rd and 4th limitation).

RELATED REVISED MANUSCRIPT TEXT:

"Third, data collection for the PEP database stopped end of 2013, which may have limited the representativeness of the PEP-seeker population presented in this study, since recent trends – such as the emergence of chemsex or the use of PrEP – could not be fully captured by our analysis. With respect to the latter, however, widespread PrEP use in Switzerland started relatively late (i.e. around

2018) compared to other countries, as supported by an online survey performed in 2017 reporting PrEP use in only 82/1893 (4.3%) of the participants. Thus, the lack of information on PrEP use may only be relevant for a small fraction of events that occurred in 2018 and later (4 out of 22 HIV events). Results generalizability may also have been hampered by fact that PEP seeking outside the USZ was difficult to assess. Although there are no surveillance data on the total of PEP prescriptions at the regional level, empirical evidence and several surveys suggest that the USZ was one of the main PEP providers in the Zurich area during this time period. Most notably, a series of longitudinal surveys among MSM found that between 2007-2014, PEP uptake ranged between 2.6 and 8% (mean: 5%). According to a previous report combining several data sources to estimate the size of local MSM populations, the number of MSM in the Zurich area was 16000 (95% credible interval: 14300 – 16500). Combining results on PEP uptake with local MSM population estimates corresponded to an estimated number of 416 to 1280 (mean: 800) PEP prescriptions occurring in Zurich within that timeframe. This suggests that the 348 MSM included in our dataset represent a substantial proportion of the total MSM population seeking PEP during the observation time. Additionally, in-depth analysis of the SHCS and ZPHI (i.e. SHCS: data on previous antiretroviral therapy (either PEP or antiretroviral therapy for HIV); ZPHI: data on previous PEP), however, did not identify PEP consultations other than those retrieved by our linkage. Although this is only valid for PEP-seekers with HIV events, it may nonetheless indicate that the USZ was one of the main PEP providers between 2007 and 2013."

MANUSCRIPT LOCATION: p.17-18, lines 21-26 and 1-17

2. Are there any potential biases in the outcome ascertainment. Are MSM more likely to be diagnosed with HIV (what is the late diagnosis rate in heterosexuals or IDUs in Switzerland)? Are MSM more likely to have been linked? Are the other groups more mobile? Less likely to be in the swiss cohort? More likely to die before testing for HIV?

Reply: this is an excellent point, thanks for bringing this up. Similar to other countries, the current HIV epidemic in Switzerland is mainly driven by MSM, who accounted for 52.7% of HIV diagnoses made in 2018, whilst transmission through heterosexual sex or IDU accounted only for 29.8% and 3.3%, respectively.⁶

You are also right to point out that risk groups differ in terms of cohort participation: evidence suggests that heterosexuals (especially infected with non-B subtypes) and IDUs are underrepresented in the SHCS cohort,^{3,7} and that heterosexuals tend to present with later stage of disease.⁸

However, it is worth to keep in mind that:

1. What applies to the SHCS would probably also apply to the PEP-USZ database. In other words, it is sensible to assume that heterosexuals reluctant to participate in the SHCS (e.g. because of low HIV/health awareness, migration background, or low trust in governmental institutions) would most likely also be reluctant to seek PEP after unprotected sex, and eventually consent to health-related data collection (as a reminder, none of the of the potentially eligible PEP-seekers records were excluded because of a veto/non-consent for data collection). Thus, hard-to-reach heterosexuals would be underrepresented both in the PEP-USZ database and the cohorts used for linkage, thereby making the risk of outcome detection bias rather unlikely.

2. As for IDUs, although we cannot formally exclude outcome ascertainment bias, the impact on our results is deemed of minor relevance, since individuals seeking PEP after sex with an IDU partner constituted only 1.1% (11/971) of the PEP-USZ database. There is also evidence that IDUs no longer contribute to the Swiss HIV epidemic, since highly efficient harm reduction programs have been in place for a long time.⁹

We adapted the limitation section to reflect these important points.

RELATED REVISED MANUSCRIPT TEXT:

"Fourth, because the HIV epidemic in Switzerland is mainly driven by MSM,⁶ and because heterosexuals tend to be underrepresented in the SHCS,^{7,8} outcome ascertainment bias may have occurred to some extent. However, this risk of bias was deemed low, since heterosexuals reluctant to participate to the SHCS would also most likely decline data collection for the PEP-USZ database, thereby leading to an underrepresentation on both sides of the linkage."

MANUSCRIPT LOCATION: p.18, lines 17-22

Abstract

1. Can the abstract follow STROBE reporting guidelines?

Reply: thanks for this suggestion. Because of the retrospective nature of data collection in the PEP-USZ database (which consists of pre-existing data, routinely collected, without pre-defined research purposes) we followed the RECORD guidelines. This is an extension to the STROBE statement, which addresses issues specific to routinely collected data that were not covered by existing reporting guidelines.

As for the specific format of the abstract, Nature Communications uses a particular format that does not include the typical sections of STROBE, CONSORT or RECORD ("The abstract should serve both as a general introduction to the topic and as a brief, non-technical summary of the main results and their implications").

RELATED REVISED MANUSCRIPT TEXT: none

MANUSCRIPT LOCATION: none

Reviewer #2 (Remarks to the Author)

This paper describes a data linkage study to estimate the long-term HIV risk among individuals seeking PEP from a tertiary setting in Switzerland. The authors find that among 975 individuals who sought PEP, 22 had evidence of HIV diagnosis from a linked cohort study, with the proportion highest among MSM (6.3%). This study provides important evidence which supports the transition from PEP to PrEP among MSM. The paper is well written and the methodology is sound.

Introduction

1. Page 5 Line 9. Is there only one recommended regimen / formulation for PEP in Switzerland. Can the authors provide a reference to the relevant PEP prescribing recommendations / guidelines?

Reply: Thanks for raising this point. The Federal Office of Public Health (FOPH) recommended indeed a standard 3-compound regimen (from 2007 to 2014: 1 protease inhibitor (such as Nelfinavir or lopinavir/ritonavir) in combination with 2 nucleoside reverse-transcriptase inhibitors (NRTIs, such as Zidovudin + Lamivudin or Tenofovir + Lamivudin or Tenofovir + Emtricitabine; as from 2014: Tenofovir/Emtricitabin + Raltegravir, with Dolutegravir or Darunavir + Ritonavir as alternatives to Raltegravir) for 4 weeks, as specified in reference #2 and 3 in our manuscript. From 2007 to 2014, the FOPH suggested alternative regimens to 1 PI + 2 NRTIs (e.g. Efavirenz + 2 NRTIs, or 3 NRTIs), but all recommended regimens were based on a 3-compound therapy (as described in the main text).

RELATED REVISED MANUSCRIPT TEXT:

We added Ref#2 and 3 at the end of this sentence (p.4 line 11), so as to improve clarity.

MANUSCRIPT LOCATION: p.4 line 11

2. Page 6 Line 9. "We assumed that HIV infections would be clustered shortly after the time of PEP consultation." It is not clear, at this point in the manuscript, what the relevance of this assumption is for the analysis.

Reply: Thanks for pointing this out. We are sorry to notice that our wording was not clear: our aim was to describe our study hypothesis, since most reporting guidelines (including STROBE or RECORD) recommend to state a prespecified study hypothesis at the end of the introduction, in the "objectives" section. We replaced "assumed" by "hypothesized" accordingly.

RELATED REVISED MANUSCRIPT TEXT:

"As previous evidence suggests that sexual risk taking occurs in phases lasting 12 to 24 months, we hypothesized that HIV infections would be clustered shortly after the time of PEP consultation."

MANUSCRIPT LOCATION: p.5, line 9

Methods

1. Page 7 Lines 10-14. Can you also describe what data is routinely collected during a PEP consultation for the PEP-USZ database?

Reply: Thanks for raising this point. We provide now a detailed overview of these routinely collected data in the online supplemental material (as reported by Marzel et al.)⁵.

RELATED REVISED MANUSCRIPT TEXT:

Online supplemental material 1: list of data routinely collected by the attending physician during PEP consultation, as reported by Marzel et al.

> Demographic data (age, sex, nationality)

> Characterization of the event leading to PEP consultation, including type of sexual intercourse [i.e. insertive, receptive, versatile (insertive and receptive), anal, vaginal, oral, smear of body fluids on healthy or wounded skin or mucous membranes, hand/feet to genitals contact, condom use, condom dysfunction], hours since exposure, or additional risk factors for HIV transmission (i.e. menstruation, ejaculation, and sexually transmitted infections)

> Result from the HIV screening test at presentation

MANUSCRIPT LOCATION: Online supplemental material 1

2. Page 7 Line 24. Did participants explicitly consent to data being linked across different studies?

Reply: Thanks for raising this important point, which has been addressed in our reply to the first reviewer (see R#1, Ethics, question #1). In brief: all participants of the PEP-database consented to the

possible further use of their health-related data for research purposes. The SHCS consent form mentions explicitly the possible further use of health-related data for research purposes and the linking of SHCS participant data to other sources of health-related data. All participants of the ZPHI retrieved by our linkage participate in the SHCS. Thus, as specified on p.7, "all provided consent, either for the use of their study data (ZPHI and SHCS cohort) or of their routinely collected clinical data (PEP-USZ database)."

RELATED REVISED MANUSCRIPT TEXT:

"This data linkage study was approved by the local ethical committee (canton of Zurich, Switzerland – Registration number: 2019-00033), as were the PEP database study (registration number: 2013-0006) and the ongoing cohorts used for data linkage (Swiss HIV Cohort Study (SHCS): registration number EK-793; Zurich Primary HIV Infection (ZPHI) study: registration number: EK-1452)."

MANUSCRIPT LOCATION: p.6, lines 3-7

3. Page 8 Line 4. By 'hospital admission' do you mean at time of admission for their PEP consultation?

Reply: yes, sorry for the misleading formulation – we adapted the manuscript accordingly.

RELATED REVISED MANUSCRIPT TEXT:

"Individuals seeking PEP for non-consensual sex or those with a positive HIV-test at hospital presentation were not considered."

MANUSCRIPT LOCATION: p.7, lines 11-12

4. Page 10 Line 16. The authors mention the possibility of repeated PEP consultations, with both consultations recorded in the PEP-USZ database. Does that mean that the 975 included PEP records refer to first-time PEP consultations only?

Reply: This is correct. The 975 included PEP records refer to 975 individuals seeking PEP between 2007 and 2013. These 975 PEP seekers generated 1051 consultations (data reported in a previous publication.⁵ We added this information in the section "Study design and setting".

RELATED REVISED MANUSCRIPT TEXT:

"(...) in brief, it consists of data from 975 individuals who sought non-occupational PEP prescription at the USZ between 2007 and 2013 (corresponding to 1051 consultations)."

MANUSCRIPT LOCATION: p.6, lines 11-13

5. Page 10 Line 22. The authors state "We assessed whether the decision to prescribe PEP was appropriate. i.e. whether the algorithm used at the time of PEP consultation was able to correctly identify individuals presenting a long-term risk of HIV infection." I am not sure whether prescribing PEP is only *appropriate* if there exists long-term risk of HIV infection. The language used by the authors implies it was only appropriate to prescribe PEP among those who eventually seroconverted. More likely, the authors sought to assess the validity or appropriateness of the algorithm (not the decision to prescribe PEP in any one instance).

Reply: Thanks a lot for highlighting this misleading formulation. You are absolutely correct, our aim was to assess the validity/ appropriateness of the algorithm. We changed the wording accordingly, and the sentence now reads:

RELATED REVISED MANUSCRIPT TEXT:

"Finally, we assessed the validity of the PEP prescription algorithm, (...)"

MANUSCRIPT LOCATION: p.10, line 13

Results

1. Page 13 Line 4: "HIV infections occurred in 2.3%..." I would change to "Evidence of HIV infection..", as you do not have HIV-negative outcomes for the rest of the cohort (some may have been diagnosed and not enrolled in any of the linked cohorts).

Reply: well noted, done as requested.

RELATED REVISED MANUSCRIPT TEXT:

"Evidence of HIV infection was found in 2.3% (22/971, 95%CI 1.48 to 3.43) of the initial PEP seekers cohort."

MANUSCRIPT LOCATION: p.12, lines 4-5

2. Page 13 Line 6. As above, I would refrain from calling those whom you do not have linked HIV diagnosis data for as "HIV-negative", as this is not confirmed.

Reply: point taken, we made the changes accordingly.

RELATED REVISED MANUSCRIPT TEXT:

1. "Compared to PEP seekers without evidence of HIV infection (...)"

2. "We also assessed the validity of the PEP prescription algorithm: compared to PEP without evidence of HIV infection, (...)"

3. Table 1: "Presumed HIV-negative"

MANUSCRIPT LOCATION: p.12, line 7; p.14, lines 20-21; table 1

3. Table 1. Please specify when age was recorded, i.e. at initial PEP consultation, or at last PEP consultation date? The latter is used to estimate median time between last PEP date and HIV diagnosis date.

Reply: Age was recorded as initial presentation, i.e. initial PEP consultation date. We adapted the table accordingly.

RELATED REVISED MANUSCRIPT TEXT:

"Age, median [IQR]; * Age recorded at initial presentation (i.e. first PEP consultation)"*

MANUSCRIPT LOCATION: table 1

4. Page 16 Line 20. As above, in your analysis of the indication for PEP algorithm, I would reconsider describing the decision to prescribe PEP as ‘appropriate’ or not.

Reply: thanks for pointing this out, we changed the wording accordingly.

RELATED REVISED MANUSCRIPT TEXT:

"We also assessed the validity of the PEP prescription algorithm: (...)"

MANUSCRIPT LOCATION: p.14, line 20

Discussion

1. Page 17 Line 8. 3.8 represents an odds ratio, so it is not correct to say a 3.8-fold increase in the *risk of HIV infection*.

Reply: thanks for pointing out this misleading formulation – we changed the wording accordingly. Of note: this paragraph has been substantially modified to meet R#1 suggestions (see "Discussion", question 1)

RELATED REVISED MANUSCRIPT TEXT:

"Clinical indicators, such as repetitive PEP seeking or indication for PEP, were associated with subsequent HIV infection."

MANUSCRIPT LOCATION: p.15, lines 5-6

2. Page 17 Lines 4-6. The authors describe that HIV diagnoses mostly occurred several years after PEP consultation, and then raise the point that “MSM should be encouraged to transition to PrEP as early as after a first PEP episode”, which seems somewhat conflicting. Perhaps the authors could elaborate on their rationale for immediate PrEP transition, given the long-term risk (1). Some further discussion around transition between periods of HIV risk would be welcomed (2).

Reply:

(1) Thanks for highlighting this aspect. The 2nd paragraph of the discussion provides now a more detailed discussion on sustained or recurrent patterns of risk over time. We also added a paragraph to the discussion that should better clarify that HIV events occurring several years after PEP consultation do in fact support fast-tracking into PrEP programs (since, as you rightly pointed out, this may reflect sustained risk over time);

(2) This new paragraph also includes a discussion on how PrEP could be an effective prevention measure in MSM with varying periods of risk.

RELATED REVISED MANUSCRIPT TEXT:

(1) "According to evidence on sexual behavior, sexual risk taking in HIV-negative MSM seems to follow distinct patterns over time: in a longitudinal analysis of data from the Multicenter AIDS Cohort Study, individuals deemed at moderate risk of seroconversion presented phases of risky behavior that lasted on average 12 months, whilst those deemed at high risk exhibited phases of 24 months. Thus, considering PEP seeking as a marker of sexual risk taking, we assumed that outcome events would be clustered 12 to 24 months after PEP seeking. Surprisingly, however, most HIV events in our study occurred several years after PEP consultation. Our study suggests thus that phases of risk may persist for longer periods than originally assumed, or that some trajectories may include recurrent

phases of risk. Large cohort studies enrolling HIV-negative individuals at considerable risk of seroconversion – such as the SwissPrEPared cohort study (NCT03893188) – will bring a better understanding of sexual behavior and risk trajectories over time."

(2) "Fast tracking PEP-seeking MSM into PrEP programs is also supported by the fact that HIV events occurred several years after PEP consultation, which suggested either sustained or recurrent risk over time. In both cases, PrEP use and – more importantly – users participation in PrEP programs could be an effective approach to provide tailored prevention measures to MSM presenting varying risk patterns over time (e.g. daily versus intermittent PrEP)."

MANUSCRIPT LOCATION: (1) p.15, lines 8-18, (2) p.16, lines 19-23

3. Given the main findings and recommendations of this paper seem to be in support of transitioning PEP users to PrEP, some discussion around how those seeking PEP can be effectively and quickly linked into PrEP programs, and be supported to remain on PrEP for long periods of time, would strengthen the discussion section.

Reply: this is an excellent point – a paragraph on that topic has been added to the discussion section.

RELATED REVISED MANUSCRIPT TEXT:

"Institutional efforts should be thus targeted at linking and retaining PEP-seeking MSM in prevention programs offering PrEP counselling, STI screening and other services related to sexual health care. To that end, approaches similar to those found effective for MSM engagement in HIV care – such as the identification of vulnerable populations, knowledge of particular sexual practices (e.g. chemsex), respectful attitude from healthcare providers, or well-defined strategies for linkage to care – should be considered."

MANUSCRIPT LOCATION: p.16, lines 23-26 and p.17, lines 1-2

4. Are the authors able to estimate an HIV incidence rate following PEP consultation, i.e. calculate person time among all PEP seekers, assuming those with no evidence of HIV diagnosis via one of the cohort studies remained HIV-negative until October 2019. This would help the reader contextualise the risk of long-term HIV among PEP seekers relative to all MSM.

Reply: Thanks for this excellent suggestion. It would be indeed very tempting to provide estimates of HIV incidence (this was actually the center of many discussions in our team). It is worth to keep in mind, however, that the interpretation of time-to-event results would require particular caution, since our study was a retrospective linkage with no systematic follow-up to ascertain the outcome (see also our reply to R#3, comment #2, which treats the exact same topic). Following the reviewers recommendation, we decided to provide a time-to-event analysis for specific subgroups of the main PEP database. These results are briefly mentioned in the discussion section (and not in the results section), so as to avoid putting too much emphasis on it.

RELATED REVISED MANUSCRIPT TEXT:

"Early transition is also supported by the fact that, compared to HIV incidence rates in the overall MSM population of Zurich (average 2010-2013: 39 per 10'000), rates retrieved from our study were higher in PEP-seeking MSM (70.5 per 10'000) and in MSM with repetitive PEP seeking (81.1 per 10'000; assuming that all HIV events were captured, and that all participants presumed HIV negative were still at risk at the time of analysis – online supplemental material 2)."

Online supplemental material 2. HIV rates in MSM seeking PEP at the University Hospital of Zurich			
Population	Nb of person-years	Nb of events	HIV rates (per 10'000 person-years)
MSM seeking PEP	3119.9	22	70.5
MSM seeking PEP, excluding possible PEP failures	3109.1	16	51.5
MSM with repetitive PEP seeking	493.2	4	81.1
MSM with repetitive PEP seeking, excluding possible PEP failures	491.9	3	61

MANUSCRIPT LOCATION: p.15-16, lines 24-26 and 1-3; online supplemental material 2.

Reviewer #3 (Remarks to the Author)

This paper examines the risk of HIV in a population based study of PEP users. While the efficacy of PEP in the short -term has been well established, long term effects and behaviour have not been well elucidated. The assessment of ongoing HIV risk is particularly pertinent given ongoing risk and the advent of effective prevention using PrEP. This clear and concise paper addresses these issues using appropriate and valid methodologies.

1. The data utilised in this study are derived from comprehensive PEP and HIV data sources. However, other than shorter duration of follow-up, it is not clear why USZ data are limited to only 2013. Is the 2013 end date to define the cohort or also data censoring (see comment below re Figure 3)?

Reply: thank you for pointing this out – this question has been partly addressed above (R#1, questions on ethics and results). Further data collection after 2013 was limited due to departmental restructuring (no data availability after this date).

The manuscript was adapted to clarify that data collection occurred retrospectively, after 2013.

RELATED REVISED MANUSCRIPT TEXT:

"Data collection occurred retrospectively (i.e. after 2013)."

MANUSCRIPT LOCATION: p.6, line 16.

2. Within the Swiss context, declining HIV incidence in MSM has been documented between 2008-2013 (<https://doi.org/10.1371/journal.pone.0131828>), and it is likely these trends have continued. It would be interesting to see if this is reflected in decreasing HIV in MSM accessing PEP, noting that event numbers are very small.

Reply: this is a very interesting point. To address this, we performed a time to event analysis among MSM diagnosed with HIV, excluding those with possible PEP failures (as this would bias the analysis), with calendar year as exposure. This analysis remained inconclusive (HR 0.94, 95%CI 0.82 to 1.08).

As mentioned in our reply to R#2 (Discussion, comment #4), it would be indeed very tempting to provide estimates of HIV incidence (this was actually the center of many discussions in our team). It is worth to keep in mind, however, that the interpretation of time-to-event results would require particular caution, since – as you rightly pointed out in your last comment – our study was a retrospective linkage with no systematic follow-up to ascertain the outcome. This may lead to an

underestimation of the number of HIV events, and to an overestimation of the number of participants considered at risk at the time of analysis.

However, following R#2 suggestions and your next comment (#3), we decided to provide a time-to-event analysis for specific subgroups of the main PEP database. These results are briefly mentioned in the discussion section (and not in the results section), so as to avoid putting too much emphasis on it.

RELATED REVISED MANUSCRIPT TEXT: see next comment

MANUSCRIPT LOCATION: see next comment

3. Additionally, including a comparison of HIV risk in this population to that in Swiss MSM, would help contextualise the conclusion that MSM using PEP group should be specifically targeted for PrEP.

Reply: this is a very good point – although we refrained from providing our main results in a time-to-event format, our paper provides now estimates of HIV incidence rates for MSM seeking PEP and MSM with repetitive PEP seeking. As discussed above, these results, however, are only briefly mentioned in the discussion section, so as to avoid putting too much emphasis on findings that require careful interpretation. As outlined in the text, the provided estimates are best interpreted as lower bounds for the real incidence in these populations. The limitation section also discusses these aspects.

RELATED REVISED MANUSCRIPT TEXT:

1. "Early transition is also supported by the fact that, compared to HIV incidence rates in the overall MSM population of Zurich (average 2010-2013: 39 per 10'000), rates retrieved from our study were higher in PEP-seeking MSM (70.5 per 10'000) and in MSM with repetitive PEP seeking (81.1 per 10'000; assuming that all HIV events were captured, and that all participants presumed HIV negative were still at risk at the time of analysis – online supplemental material 2)."

2. "A time-to-event analysis was conducted to explore HIV rates in some specific subgroups (online supplemental material 2). These findings, however, need to be interpreted with caution, since the lack of systematic follow-up for outcome ascertainment may have led to an underestimation of the number of HIV events, and to an overestimation of the number of participants considered at risk at the time of analysis. Hence the provided incidence rate estimates are best interpreted as lower bounds for the real incidence in these populations."

Online supplemental material 2. HIV rates in MSM seeking PEP at the University Hospital of Zurich			
Population	Nb of person-years	Nb of events	HIV rates (per 10'000 person-years)
MSM seeking PEP	3119.9	22	70.5
MSM seeking PEP, excluding possible PEP failures	3109.1	16	51.5
MSM with repetitive PEP seeking	493.2	4	81.1
MSM with repetitive PEP seeking, excluding possible PEP failures	491.9	3	61

MANUSCRIPT LOCATION: p.15-16, lines 24-26 and lines 1-3; p.19, lines 1-6; online material 2

The data linkage methodology used in this study is the most relevant for this type of study question. The description of linkage process is clear and sufficient. The ability to assess HIV linkages through review of clinical records and eliminate false positives is a strength of this study. The simulation methodology to assess false negatives could be expanded in the methods.

Reply: thanks for this comment, we added this information in the methods section.

RELATED REVISED MANUSCRIPT TEXT:

"Finally, to estimate the expected false-negative proportion yielded by the linkage method, we performed a simulation study using two pre-existing registries (HIV data consisting of 2,868 records, and Cancer Registry data consisting of 90,276 records). The simulation linkage between these registries used the exact same variables as those used in our study. The proportion of false negatives was estimated by comparing the results of the simulation linkage study with a gold-standard, i.e. with the results from a previous linkage performed between these registries using several highly discriminative variables (such as date of birth, date of death, names, or place of residence)."

MANUSCRIPT LOCATION: p.9-10, lines 20-25 and 1-2

It would also be beneficial to include uncertainty bounds for the assessment of HIV risk in this population.

Reply: excellent suggestion, we computed 95%CI for the overall and MSM PEP-seeker population and added this information in the relevant sections.

RELATED REVISED MANUSCRIPT TEXT:

- 1. "For the assessment of HIV events, proportions with their 95% confidence intervals (CI) were computed."*
- 2. "Evidence of HIV infection was found in 2.3% (22/971, 95%CI 1.48 to 3.43)."*
- 3. "When only MSM were considered, the proportion of individuals with evidence of HIV infection reached 6.3% (22/348, 95%CI 4.17 to 9.43)."*

MANUSCRIPT LOCATION: p.10, lines 9-10; p.12, lines 4-5; p.12, lines 9-10.

While the usual method of analysis for this type of data would be time to event, the lack of routine screening for events, along with the other reasons described by the authors, supports summary of events as risk.

Specific comments:

Pg 16, Ln 19: for clarity "thereby impairing meaningful multivariable regression"

Reply: thanks for pointing this out, the wording has been corrected accordingly.

RELATED REVISED MANUSCRIPT TEXT:

"(...), thereby impairing meaningful multivariable regression analysis."

MANUSCRIPT LOCATION: p.14, line 18.

Pg 16 In 24: does this include adjustment for PEP post 2013?

Reply: this question has been partly answered above, in reply to your first comment. All adjusted analyses refer to the period 2007-2013, as no data were available after 2013. It is however good to remember, that both the SHCS and ZPHI collect information on PEP episodes prior to HIV infection. Although this only applies to PEP-seekers diagnosed with HIV, in-depth review of the SHCS and ZPHI data did not identify PEP consultations other than those retrieved by our linkage (as mentioned in the first version of this manuscript).

RELATED REVISED MANUSCRIPT TEXT: none
MANUSCRIPT LOCATION: p.14, lines 12-13

Pg 26, Figure 3. Were there any repeat occurrences of PEP between 2013 and 2018?

Reply: please refer to our reply to your first comment and to the comment above ("Pg 16 In 24: does this include adjustment for PEP post 2013?"). All analyses refer to the period 2007-2013, as no data were available after 2013.

RELATED REVISED MANUSCRIPT TEXT: none
MANUSCRIPT LOCATION: none

Professor Janaki Amin
Macquarie University, Australia

REFERENCES

- 1 Locicero, S. & Bize, R. Les comportements face au VIH/Sida des hommes qui ont des rapports sexuels avec des hommes - Enquête Gaysurvey 2014 (Institut universitaire de médecine sociale et préventive, Lausanne, 2015).
- 2 Schmidt, A. J. & Altpeter, E. The Denominator problem: estimating the size of local populations of men-who-have-sex-with-men and rates of HIV and other STIs in Switzerland. *Sex Transm Infect* **95**, 285-291, doi:10.1136/sextrans-2017-053363 (2019).
- 3 Thierfelder, C. *et al.* Participation, characteristics and retention rates of HIV-positive immigrants in the Swiss HIV Cohort Study. *HIV Med* **13**, 118-126, doi:10.1111/j.1468-1293.2011.00949.x (2012).
- 4 Hampel, B. *et al.* Assessing the need for a pre-exposure prophylaxis programme using the social media app Grindr(R). *HIV Med* **18**, 772-776, doi:10.1111/hiv.12521 (2017).
- 5 Marzel, A. *et al.* Prescription of Postexposure Prophylaxis for HIV-1 in the Emergency Room: Correct Transmission Risk Assessment Remains Challenging. *J Acquir Immune Defic Syndr* **74**, 359-366, doi:10.1097/QAI.0000000000001265 (2017).
- 6 Bundesamt für Gesundheit. HIV, Syphilis, Gonorrhoe und Chlamydiose in der Schweiz im Jahr 2018: eine epidemiologische Übersicht. *BAG-Bulletin* **41**, 10-36 (2019).
- 7 Shilaih, M. *et al.* Genotypic Resistance Tests Sequences Reveal the Role of Marginalized Populations in HIV-1 Transmission in Switzerland. *Sci Rep* **6**, 27580, doi:10.1038/srep27580 (2016).

- 8 Buetikofer, S., Wandeler, G., Kouyos, R., Weber, R. & Ledergerber, B. Prevalence and risk factors of late presentation for HIV diagnosis and care in a tertiary referral centre in Switzerland. *Swiss Med Wkly* **144**, w13961, doi:10.4414/smw.2014.13961 (2014).
- 9 Marzel, A. *et al.* The Cumulative Impact of Harm Reduction on the Swiss HIV Epidemic: Cohort Study, Mathematical Model, and Phylogenetic Analysis. *Open Forum Infect Dis* **5**, ofy078, doi:10.1093/ofid/ofy078 (2018).

REVIEWERS' COMMENTS

Reviewer #1 (Remarks to the Author):

The Authors have done an excellent job of responding to the many reviewer comments, including mine. I have no further issues to raise and am happy with the response and revised manuscript

Reviewer #2 (Remarks to the Author):

The authors have addressed all my comments and suggestions satisfactorily. This is an important and interesting paper.

Reviewer #3 (Remarks to the Author):

I consider all reviewers comments to be sufficiently responded to and appropriately incorporated in to the revised manuscript.